# Jonah in 20th Century Literature

**Lena-Sofia Tiemeyer**

Theology Department, Örebro School of Theology, Box 1623, 701 16 Örebro, Sweden;
lena-sofia.tiemeyer@altutbildning.se

**Abstract:** The biblical book of Jonah has been the subject of multiple literary retellings, ranging from individual poems to whole novels and theatrical dramas. This article focuses on interaction with the book of Jonah in 20th-century world literature, where Jonah becomes our alter ego; he embodies our own struggles with God. I shall highlight three common tropes in the retellings: (1) Several retellings use the character of Jonah to express a person's failure to escape God's calling. Others use him to explore the Jewish experience of never being able to run away from being chosen by God. (2) Other retellings turn the trope of "the fleeing Jonah" into "Jonah the refugee": Jonah is a man whom God abandoned. These retellings stem from Jonah 2:5 (Eng. 2:4] where Jonah expresses how he is cast out from God's presence. They gain further inspiration from the affinity between the dialogue between God and Jonah in Jonah 4 and that between Cain and God in Gen 4. This intertextuality fashions Jonah as a type for the "wandering Jew." (3) Yet another set of retellings employs the figure of Jonah to discuss God's justice and his perceived failure to be unmerciful.

**Keywords:** Jonah; literature; poetry; novels; reception history; reception exegesis; history of interpretation; Judaism; Hebrew prophets; prophetic literature; ancient Israel

## 1. Introduction

The biblical book of Jonah has been the subject of multiple literary retellings, ranging from individual poems to whole novels and theatrical dramas. In addition, a huge number of literary works allude to the Jonah narrative in more or less explicit ways. In the present article, I shall limit the focus to the interactions with the book of Jonah that are found in twentieth-century world literature. We shall discover that Jonah has often become a representative of humanity. Jonah is haunted by God, exiled from his land, and persecuted for his beliefs. In many respects, Jonah is presented in the literature as our alter ego, who embodies our own struggles with God.

## 2. Methodology

This article falls methodologically in the realm of "reception exegesis", a term coined by Paul Joyce and Diana Lipton in their *Wiley Blackwell Commentary to Lamentations* (Joyce and Lipton 2013). Whereas *Rezeptionsgeschichte* ("reception history") and *Wirkungsgeschichte* ("impact history") explore how the biblical text has been conceptualized and interpreted, and how these interpretations have influenced our ways of thinking and acting, "reception exegesis" puts the focus squarely back on the biblical text; it investigates how the reception history of a given text helps us to see aspects of that same text that are present, yet often hidden from plain sight due to our own predetermined ideas of what the text ought to be about (Joyce and Lipton 2013, pp. 17–19). In my view, reception exegesis can enhance our reading of the book of Jonah as it draws our attention to aspects of the biblical texts that, although noted by exegetes, rarely are afforded any lengthy treatment.

Inspired by this approach, I shall use modern fiction as a lens through which to read the book of Jonah. In short, how do these contemporary authors bring out into the open matters that lie dormant in the text? Yet, rather than offering a systematic application of reception exegesis on the book of Jonah, my aims here are more modest. By analyzing the

interaction between this biblical book and twentieth-century creative writings and how modern authors have used biblical themes and motifs artistically to add depth to their writing, I hope to provide glimpses of the interpretative insights that such meetings generate. More personally, I shall discuss how, in several instances, a poem or a novel has encouraged me to explore a hitherto neglected interpretative trajectory. To emphasize the exegetical aspects of my investigation, whereby the fictional works *interpret* the biblical narrative, I shall label the novels and poems under discussion "retelling" or "re-reading." Retellings have the biblical text as their clear point of departure, constant fixture, and explicit aim.

To set the stage, a (biblical) text does not have a single, stable, and determinate meaning. Rather, meaning is supplied to the reader through an interactive and dynamic process. In this manner, every act of reading involves interpretation, and meaning originates in the reader's response to the text. On the one hand, the text itself sets certain boundaries which preclude a text from meaning whatever the reader wants it to mean; on the other hand, the reader fills the gaps in the text and thus imbues it with meaning (Iser 1978, pp. 274–94). Yet, the determination of the meaning that a text can and cannot have is often less a matter for each reader and more a matter for the interpreting community (Fish 1980, pp. 338–55): it is not merely each individual reader's fantasy that determines how the gaps of the text are filled, it is also the case that the norms of the surrounding society direct readers towards filling them in distinct and (from their perspective) acceptable ways. The result is a dialogue between the text and the readers: although the authors with whom I shall converse use the biblical narrative as their starting point, they allow their subjective concerns and those of the people around them to shape their understanding of the text.

My reception exegetical approach here enables me to treat contemporary fiction as a form of midrash. Midrash in its most narrow sense is an ancient Jewish commentary. It seeks to resolve the interpretative problems posed by difficult passages in the Bible. Midrash responds to two interrelated problems. First, as articulated by Harold Fisch, the biblical texts are characterized by narrative gaps, both in terms of characterization and the sequence of events. Midrash responds to these "missing" elements: it clarifies and, when needed, also adjusts the plot progression, providing the characters with motives and emotions, while creating a dialogic "give-and-take." In this respect, midrash finds itself between interpretation and invention. Secondly, the biblical narrative itself rarely offers guidance on how to evaluate what happens, and seldom tells us what to think. As a result of this authorial silence at critical junctures, the text becomes polyphonic. It draws the reader in and compels the search for a hidden dimension, which will inevitably differ from reader to reader (Fisch 1998, p. 17). As a result, the message of a biblical text becomes aligned with the interpreter's own ethical and religious values and expectations. As such, in line with Fish's notion of an interpretative community, midrash interprets and completes the biblical text in accordance with its cultural codes (Boyarin 1990, pp. 11–14). At the same time, midrash, understood as religious texts, establishes boundaries for what the audience is allowed to hear from the text and thus ultimately reinforces the extant cultural codes.

In a much broader sense, the term midrash is used in studies that explore the literary afterlife of the Bible. Notably, Lesleigh Cushing Stahlberg employs it to denote any type of literature which interacts with and interprets the biblical material in the above-mentioned manners and for the above-mentioned purposes (Cushing Stahlberg 2008, pp. 92–135; 2009, p. 33). Likewise, Fisch argues that modern literature can be seen as an extension of the Midrashic mode, whereby the act of reading is combined with "the fertile play of imagination" (Fisch 1998, pp. 17–18). At the same time, the biblical text exercises a certain constraint, as it hovers constantly in the background and asserts its authority. The biblical text remains, and every single new retelling exists in sustained dialogue with it (Fisch 1998, pp. 19–20). Biblical retellings are thus inherently exegetical endeavors (cf. Fisch 1998, p. 25).

The present article showcases how a wide range of twentieth-century novels, short stories, and poems function as this broader form of midrash. Contemporary literature, to paraphrase the words of Vladimir Tumanov, may yield insight into the meaning(s) of the text itself as it challenges readers to leave their traditional assumptions behind and instead pushes them to detect hitherto unseen nuances in the text (Tumanov 2009, p. 509). In the specific case of the book of Jonah, the cited literary works encourage readers to reflect on aspects of the biblical text which, although commented on by exegetes, seldom are given pride of place in scholarly discussions.

Even though the individual authors responsible for these fictional works probably would not regard themselves as biblical exegetes, they are undoubtedly aware of the interpretative quality of their work: by choosing a (biblical) text as the starting point and source of inspiration, the result will inevitably constitute some form of midrash according to the wide definition advocated here. To cite Robert Alter, speaking of the choice of modern authors as conversational partners when doing exegesis, "the canonicity of the Bible was acutely palpable to them, imaginatively available to them, perhaps in some ways that it might not be to the conventional believer. The inventive and at times disorienting use of the Bible in their writing is a vivid manifestation of the dialectic of iconoclasm and traditionalism that informs a good deal of modern writing" (Alter 2000, p. 9). A creative writer may in some cases be more attuned than biblical scholars to the different literary nuances that a text can have, and biblical scholars do well to listen to their expertise.

Fictional retellings do not only shed new light on the biblical texts but also subvert them by stressing those nuances that create a sense of unease. As such, retellings challenge their audience to confront such matters of tension and may even compel them to reject the claims of the text. Fisch, in his discussion of the influence that the Bible wields over (western) literature, highlights how this influence not only has a shaping character but, at the same time, is "profoundly antithetical": the biblical stories are not only echoed but also resisted, inverted, and satirized. "The western imagination cannot escape it but neither can it accept it unaltered" (Fisch 1998, p. viii). Piero Boitani likewise shows how later authors, intrigued by the biblical narratives, cannot let them be. Instead, they keep rewriting them, developing them, and giving them closure, while also allowing the biblical stories to resonate against other known texts and thus to inform and to be informed by them in a truly intertextual web. The biblical characters are rewritten to resonate with us, their words and actions are fleshed out as they echo the words and actions of other characters, both within the Bible and in other literature. Retellings, furthermore, often form critical interpretations. The authors, by twisting the plot or exploring the openness or ambiguity of the text, question and problematize the decisions that the biblical characters make. Finally, retellings are free to foreground an event, remove parts of the biblical text, or narrate the sequence of events in a different order, thereby shifting its emphasis and destabilizing its message (Boitani 1999, pp. 33–37).

All literary retellings (as well as more pictographic ones, such as those found in paintings and sculpture) represent interpretations of the Bible and, like all interpretations, form two-way movements. In one direction, the authors allow their assumptions about the narrative to influence their retelling. The biblical narratives are seldom explicit about a character's feelings, for example, and retellings duly fill these blanks to satisfy their readers' need to know what the dramatis personae felt in a given situation. In this process, biblical characters are aligned with the readers' expectations and become subject to their (moral) evaluation. Their presuppositions about what the biblical text is about, or, rather, what it should be about, are consciously or subconsciously read into the narrative. In the other direction, the retellings shape and sometimes even fundamentally alter how we understand the biblical text. When we pick up the Bible the next time, the images from the retellings linger in our minds and impact our appreciation of the biblical text. In short, literary retellings not only reflect the views of (popular) culture but also cement those same views in its memory (Exum 1996, pp. 62–64; cf. Kreitzer 2002, pp. 8–9). The dialogic character of retellings, cre-

ated by the encounter between the biblical text and the retelling, is an active relationship that influences in both directions (cf. Fisch 1998 above).

## 3. Retellings of the Jonah Narrative

In the rest of this article, I shall explore how literary retellings of the book of Jonah function in the above-mentioned capacity of shaping, subverting, and problematizing its message(s). There are three common tropes in the retellings: Jonah who runs away from his calling, Jonah the refugee, and Jonah who questions God's justice. I shall argue that the dialogue between text and retelling enhances our appreciation of the biblical text by highlighting extant nuances in the text that often remain peripheral in "conventional" biblical exegesis.

### 3.1. Jonah's Flight

Many scholars through the ages have pondered the issue of Jonah's flight from God's command in Jonah 1:3 (see my survey in Tiemeyer 2022, pp. 29–48, with cited bibliography). The biblical narrative provides one explicit answer: Jonah, due to his prophetic office, knew that Nineveh would repent, a notion that is explicated later in Jonah 4:2. Expressed differently, the statement in Jonah 4:2 is thus read back into the fabric of Jonah 1:3 and understood to provide the key to Jonah's disobedience. Many modern poets and novelists agree. In the present context, we shall investigate two sets of tropes in modern literature that have their origin in Jonah's flight in Jonah 1:3 and the presumed reason for it in Jonah 4:2.

Beginning with Jonah 1:3, we shall look at how modern novelists and poets have utilized the character of Jonah to explore the topos of fleeing from one's calling. We shall further discuss how Jonah has come to represent the notion of seeking to escape from one's destiny, either generally as a human being or more particularly as a Jew. Finally, from the opposite perspective, we shall also note how in a few cases Jonah serves as a lens for probing the Jewish experience of rootlessness, exile, and a sense of being banished by God. Turning to Jonah 4:2, we shall examine how modern novelists and poets have used Jonah to discuss issues of divine justice and what can most aptly be called God's perceived failure to be unmerciful. Jonah voices our own outcry at the discrepancy between crime and punishment and our discomfort in the face of cheap mercy.

### 3.1.1. Fleeing from One's God-Given Calling

The theme of Jonah's attempted flight from his God-given task of declaring God's doom over the people of Nineveh (Jonah 1:3), long noted by not only traditional Jewish and Christian exegetes but also modern critical scholarship, has been picked up in creative ways by many twentieth-century novelists and poets and made to resonate with modern life. Their literary retellings transform and, to a large extent also subvert, Jonah's ultimately failed escape by positioning it within the wider framework of humanity's relationship with God. It has also been understood more specifically within the context of, on the one hand, God's covenant with Israel and, on the other hand, the new covenant between God and the Church. Common to many of these retellings is the theme of futility and frustration. The Jonah narrative is evoked to emphasize that all attempts to escape one's God-ordained fate will ultimately fail. The dialogue between literature and the biblical text thus becomes the vehicle for depicting the modern struggles to evade one's destiny.

A good example is the novel *Jonah's Gourd Vine* (Hurston 1934) by the African American novelist and Baptist preacher Zora Neale Hurston (1891–1960). The protagonist, the preacher John Buddy Pearson, is a modern Jonah. Like Jonah, he is commissioned by God to speak not his own words but God's; like Jonah, he initially resists God's words; and finally, like Jonah, he fails to internalize God's message in his own life. Both narratives end without their protagonists having reached anything akin to an understanding of God's ways. At the end of the biblical story, Jonah is left sitting outside Nineveh with no sign of having accepted God's viewpoint. Likewise, at the end of Hurston's book, Pearson is

killed by a train without fully understanding what his life was all about. Pearson, like Jonah, ultimately refuses to allow God to transform him (Ciuba 2000, pp. 126–27, 130).

Another example is the novel *Guldspiken* (*The Golden Nail* (Nilson 1985)) by the Swedish author Peter Nilson (1937–1998). The novel's dialogue with the book of Jonah explores a person's feeling of despondency when faced with the inability of escaping his/her God-ordained fate. The main protagonist, the poor boy Elias, is led to believe that he is called to travel to the glorious land of opportunities, America, where the railroad is nailed with golden nails. This dream is interrupted, though, by a voice that calls him to go and preach in hell. To escape his calling, Elias flees on a ship which takes him to Africa, only to realize that life aboard is truly hell on earth. His flight from his calling has done nothing but bring him to the place where he did not want to go in the first place. God summarizes his situation aptly for him upon his return to Sweden: "No person ever escapes her fate" (my translation) (pp. 98–100, 109–33).

Hurston's and Nilson's interactions with the book of Jonah create a two-way movement. On the one hand, the dialogue with the biblical text deepens the readers' impression of the futility of John Buddy Pearson's and Elias's endeavors: we all know that the biblical Jonah fails in his attempt and so we anticipate that the protagonists of the two novels will do the same. The biblical echoes thus add a deeper dimension to the literary retelling. On the other hand, the retellings challenge our appreciation of the book of Jonah or, expressed differently, they open the door for us to access a set of interpretations of the biblical book that is more attuned to our modern sensibilities and that resonate with our own questions. The biblical prophet, presented as a caricature due to his wilful and grumpy manner, is thus transformed into a man of flesh and blood that modern readers can relate to and for whom we feel compassion. He is trapped by God and, despite his best efforts, is forced to do something that he does not wish to do. Jonah, read through this lens, becomes a victim and God becomes a bully.

### 3.1.2. Humanity's Alienation from God

Other retellings stress more broadly Jonah's alienation from God and the world. The recalcitrant biblical prophet and his ensuing time inside the fish are turned into a symbol of the pan-human or, more specifically, Jewish struggle with God. As Yvonne Sherwood aptly states, Jonah becomes "a patron saint of all those who feel the need to curse" due to the unsettledness and fickleness of their existence (Sherwood 2000, p. 171).

Paul Auster (b. 1947), for example, employs the motifs of "being inside the whale" and of "shipwreck" as leitmotifs throughout his book *The Invention of Solitude* (Auster 1982) to designate the estrangement that characterizes much of post-holocaust Jewry. The character George Oppen uses the term "shipwreck of the singular" to refer to his own sense of being separated and even banished from the rest of humanity (p. 83, cf. p. 96). Oppen likens his own existence to Jonah running away from the presence of the Lord to meet his doom by shipwreck. In Auster's hands, the story of Jonah becomes one of solitude and Jonah becomes a prophet of silence:

> If the Ninevites were spared, would this not make Jonah's prophecy false? Would he not, then, be a false prophet? Hence the paradox at the heart of the book: the prophecy would remain true only if he does not utter it. But then, of course, there would be no prophecy, and Jonah would no longer be a prophet. But better to be no prophet at all than to be a false prophet "Therefore now, O lord, take, I beseech thee, my life from me; for it is better for me to die than to live." Therefore, Jonah held his tongue. Therefore, Jonah ran away from the presence of the Lord and met the doom of shipwreck. That is to say, the shipwreck of the singular.

Jonah's struggle with God is also the topic of several poems. The American Jewish poet Gabriel Preil (1911–1993), originally from Estonia, compares himself to Jonah, and uses the biblical character to describe an existence torn between faith and a desire to flee from it (Preil 1961):

> *The prophet Jonah ran from his angry Master*
> *And I to my ship empty of God and man*
> *[ . . . ]*
> *I, God willing, while escaping my Master, hope to find*
> *A minute of refuge in a season of faith and ripeness.*

The same sense of struggle with God permeates a poem by the Chilean poet Enrique Lihn (1929–1988), who uses the book of Jonah as a lens through which to express his unease with life. In this poem, Jonah contemplates the fickleness of not only his own existence but also that of God. As expressed by Sherwood, Jonah observes how God is participating in his own erosion, as he wavers "between mercy and anger" (Lihn 1972; Sherwood 2000, p. 192):

> *In the name of Isaiah, the prophet, yet with the grotesque and unfinished gesture of his*
> *colleague Jonah,*
> *who never managed to get through with his simple task, given to the ups and downs*
> *of good and evil, to the fickle circumstances of history that plunged him into the*
> *whale's belly.*
> *[ . . . ]*
> *And Jehova's doubts about him, wavering between mercy and anger, grabbing him and*
> *tossing him that old instrument whose use is doubtful*
> *no longer used at all any more.*

The American-born Israeli poet Ariel Zinder (1973–) likewise evokes the book of Jonah. The allusion is explicated by the title of his first book of poetry, *The Ships of Tarshish*, a title which simultaneously recalls the wealthy merchant ships traversing the Mediterranean with valuable stock and their downfall (Isa 23:14). His poem "*To the Weak, the Fearful and the Faint-hearted*" (Zinder 2007) features the prophet Jonah as its protagonist. The poem, located in Jaffa (Jonah 1:3), depicts how Jonah is running around the port trying to flee from himself. He is weak and despairing, and he both fears and shuns the society around him. The poem ends with the contrasting image of Jonah's demise in the watery abyss, contrasted with the idealized ports of Jaffa with its "golden alleys":

> *Jonah goes running through the alleys. The ship is already at anchor, the quayside stir-*
> *ring to life.*
> *You who gather there, hands outstretched, hearts exposed—leave him alone. Don't*
> *mock him.*
> *For even if you hold out a begging-bowl, he'll scorn the wretched tremor in your voice.*
> *Turn away. He is weak, he's shaking. Let him hurl himself towards Tarshish.*
> *Your gaping wounds revolt him. If you ask, he will say you've no strength for a cure.*
> *He will say he hears the voice of mercy, and yet will not submit to it.*

A subset of these retellings deals with the Jewish experience of never being able to run away from being chosen by God. The notion of the Jewish people carrying a burden and having a responsibility towards God and towards the Gentile world is expressed poignantly by the Russian-Jewish author Kadia Molodowsky (1894–1975) in her poem "*Jonah*" (Molodowsky 1965). Mixing the image of the Jews being marked for God (Gen 4:15, see below) and thus unable to escape their destiny, with images of persecution, sickness, and abuse, it expresses powerfully the Jewish experience through the lens of Jonah's calling. Jonah the Jew can never escape from being God's chosen people:

> *[ . . . ]*
> *Lamenting, you will plead with God:*
> *Why am I your vessel, why me?*
> *I want to plant a date palm, an apple tree,*
> *[ . . . ]*
> *And a voice answers you from the storm:*
> *Forget your apple tree, your house and your kin,*
> *You are chosen for mercy and for pain,*

*Go to Nineveh,*
*And purify its sin.*

As in the cases above, the sustained dialogue of these authors with the Bible creates a two-way movement that not only provides depth to the readers' appreciation of the novels and poems in question but also influences our perception of the biblical character of Jonah and our understanding of the overarching message of the biblical book bearing his name.

Focusing on the latter issue, the way in which the modern, post-holocaust crisis is being read into the fabric of the biblical text challenges Bible-readers to reevaluate Jonah's failure to flee from God's calling. We are encouraged to ask tough questions about what it means to be unable to flee God's selection and to explore the torment that being God's chosen people entails. Read this way, the biblical Jonah becomes the victim of God's relentless insistence that he obeys his commands. Further, the psalmist's words about God's care in Ps 139:7–12 become a taunt rather than a blessing.

In addition, the dialogue between literature and the biblical text encourages the readers to identify with Jonah—like the biblical character, we cannot hide from God's eyes—and thus to deconstruct the overt message of mercy in the biblical book. Rather than accepting God's take on the situation, we are provoked into contemplating alternative viewpoints and to wonder whether Jonah's attempted flight from his God-given task was justified after all.

Finally, when the book of Jonah is being read through the lens of these novels and poems, we are forced to reconsider its genre (see further 3.2.1). It becomes almost untenable to read it as a humorous book about a man who is swallowed by a fish. Rather, these literary retellings dare the readers to take the book of Jonah with utmost sincerity and approach it with fear and trepidation.

### 3.1.3. Jonah the Refugee and the Perpetual Exile

Yet other, related retellings turn the trope of "the fleeing Jonah" into "Jonah the refugee" and "Jonah the exile": Jonah is a man whom God abandoned. As I have discussed in an earlier article, these retellings draw inspiration from the shared use of four expressions, the verb "banish" (root גרשׁ), the verb "to be angry" (root חרה), the expression "from before the face of YHWH" (מלפני ה'), and the expression "east" (מקדם), which are interwoven into the textual fabric of both Jonah and Gen 1–4 (Tiemeyer 2019; 2022, pp. 215–17, cf. Berger 2016, pp. 13–14).

In the book of Jonah, these four expressions are attested in Jonah 1:3, 10 where Jonah flees from "before the face of YHWH" (מלפני ה'), in Jonah 2:5 (Eng. 2:4]) where Jonah describes how he is "cast out" (נגרשׁתי מנגד עיניך) from God's presence, in Jonah 4:1, 4 which expresses Jonah's "anger" and God's quest for an explanation of said anger (v. 1, וירע אל יונה רעה גדולה ויחר לו, v. 4, ויאמר ה' ההיטב חרה לך), and in Jonah 4:5a where Jonah leaves from Nineveh to "the east" (מקדם) of the city.

Together, these expressions link the book of Jonah to the narrative about Cain in Gen 4. Cain is "angry" with God for not accepting his sacrifice (Gen 4:5bα, ויחר לקין מאד, and 4:6bα, למה חרה לך) and kills his brother Abel. Afterwards, God declares that Cain will be a restless wanderer on the earth (v. 12). In response, Cain states that God has "driven him away" from the face of the earth (v. 14a, הן גרשת אתי היום מעל פני האדמה). To keep him safe, but also as a constant reminder of his heinous deed, God puts a mark on Cain. Cain ultimately sets out "from YHWH's presence" (v. 16a, ויצא קין מלפני ה') and ends up "east of Eden" (וישב בארץ נוד קדמת עדן ויצא קין מלפני ה', cf. also Gen 3:24, where God drives out Adam and Eve "east" of Eden). When reading Jonah together with Gen 4, an evocative portrait of Jonah as a type for Cain and accordingly for the "wandering Jew" takes shape.

The understanding of Jonah as a type for the perpetually rootless Jew comes to the forefront in the novel *The Strange Nation of Rafael Mendes* by the Jewish Brazilian author Moacyr Scliar (1935–2011). The Jonah narrative lends structure to the novel (Scliar 1987).

*I was able to track down someone named Jonah as the most distant of my known ancestors.*
*[ . . . ] Throughout the ages, they fled from place to place, from country to country.* (p. 75)

The main character Rafael appears, again and again, in different incarnations. As the novel progresses, the reader encounters a long sequence of Jonah's descendants, each of them characterized by their rootlessness. Habacuc—a Jonah/Rafael Mendes reincarnation living in the first century CE—flees from the Romans to Yafo, boards a ship, and ends up in Spain (presumably an allusion to Tarshish). As his flight coincides with the death of Jesus, Habacuc's descendants are doomed to be rootless:

> *"Your crime didn't go unnoticed", replied the voice. "Because of the sins you and others have committed, a god has died, Habacuc. As punishment, your descendants will wander the earth until they finally hark to the word of the children of light. Have I made myself understood?".* (p. 93)

Rafael Mendes, who lives during the Inquisition, is a New Christian (cristãos-novos). After imprisonment and prolonged torture, Rafael and his companion Afonso manage to escape, only to end up on a ship run by Jew-hating sailors. Near the coast of Brazil, the weather suddenly changes. Having found out that Rafael and Afonso are Jews (cf. Jonah 1:9), the sailors turn against them and hold the two "descendants of Christ's killers" responsible for the storm:

> *"Divine punishment has befallen us", muttered the sailors, "for we are harbouring two heretics, two descendants of Christ's killers." Tension kept mounting, and one night Rafael and Afonso woke up with shouts and the clangor of swords. [ . . . ] "Save yourselves", the captain shouted at them, "jump into the sea".* (p. 126)

The motif of the sailors' act of throwing Jonah overboard to save themselves (v. 15), somewhat justified in the biblical narrative, is here subverted into a breach of hospitality and wilful murder of the already persecuted Jewish men. Rafael, like Jonah, survives the ordeal, but stays homeless and continues his restless wandering of the world.

In the penultimate incarnation, Rafael Mendes seeks to return to Spain to fight with the Republicans. He does not reach Spain, however, but dies aboard the ship and, like Jonah, his body is thrown into the sea:

> *During a brief moment of lucidity, he announced that he would be dying soon; he asked that his body be cast into the sea so that like Jonah (to use his own words) he could reach his destination.* (p. 259)

In Rafael, Scliar has created an alternative type of Jonah. Rafael, like Jonah, flees but, unlike Jonah, he lacks agency. Whereas Jonah initiates his flight in rebellion against God, Rafael's flight is merely a response to all the events that befall him (Barr 1996, pp. 42–45). Yet, the lasting impression of both men is that of the eternal exile who, haunted by God, is bound to be homeless and constantly searching for a place to rest.

The related notion of Jonah as the eternal exile is epitomized in the short story "*My Christina*" ("La meva Cristina") by the Catalan novelist Mercè Rodoreda (1908–1983) (Rodoreda 1984). She uses the Jonah narrative as a lens through which to explore the notion of marginalization and estrangement that are the companions of exile. A shipwrecked sailor is swallowed and thus saved by a whale. At first, the sailor is grateful to the whale and gives her the name "Christina", named after the boat that has been shipwrecked and that he has been forced to abandon. Halfway through the narrative, however, his relationship with the whale changes. Realizing that she is his prison, he turns against her. Yet, as time goes by, his attempts to hurt her become more and more feeble, and he begins to accept his destiny of never being able to leave her. The whale, from her perspective, covers the sailor in layers of mucus that eventually form a shell, like a pearl, to render harmless this foreign object in her body and thus to save herself from his destructive actions. Much later, after the death of the whale, the sailor is free to return home. At that point, he is changed beyond recognition. As a result, he is unable to reintegrate into the home that he had left so many years ago, and his former compatriots are unable to accept him as the one he has become. The fish, as personified "exile", first perceived as a haven and later as a prison, has turned the sailor into a hybrid that does not fit anywhere anymore (Nichols 1986, pp. 405–17).

The dialogue between the book of Jonah, on the one hand, and the works by Scliar and Rodoreda, on the other, exemplifies in many ways the fruit of reception exegesis. From a diachronic perspective, these two literary works prompted me to look for and subsequently also to discover the aforementioned intertextual links between the book of Jonah and Gen 1–4. Prior to reading Scliar's and Rodoreda's works, I was unaware of the connection between the two biblical texts; after reading them, my understanding of especially Jonah 4 was forever changed. The open-ended image of the prophet, sitting east of Nineveh, at the end of the chapter, not knowing whether to stay in Nineveh or go home, brought a poignancy to the book for which I was unprepared. Has Jonah forfeited the right to his home after having ensured Nineveh's survival? At the same time, to what extent has Jonah lost his prophetic profession by his flight from God's command? Like the sailor in Rodoreda's short story, Jonah has not only failed as a refugee, but he has also lost his ability to return home. Furthermore, like Scliar's Jonah, saving the Ninevites has not made him, an Israelite, one of them. He is still a foreigner in Nineveh, and it is still his homeland that the Neo-Assyrians are going to destroy in a mere few years' time (see further below).

### 3.2. God's (Lack of) Justice

This brings us to the reasons for Jonah's flight from God in the first place. In the biblical narrative, Jonah's disobedience is explained in Jonah 4:2—Jonah knew that God is a God who is prone to showing compassion. The question then becomes: why did Jonah not wish God to show compassion for Nineveh? From the opposite perspective, why did God decide to extend his mercy to Nineveh, despite its attested cruelty against the surrounding nations (Nah 3:19) and its destructive acts against the people of Israel (2 Kgs 17)?

### 3.2.1. God's Failure to Be Unmerciful

Several modern literary retellings of the Jonah narrative pick up these questions and use the Jonah narrative as a lens through which to ponder God's mercy or, more commonly, his failure to withhold mercy. The pivotal issue is the balance between mercy and justice: can true mercy exist in the absence of true justice? Put aptly by the British literary theorist Terry Eagleton (1943–), Jonah refuses to obey God because there does not seem to be a point in obeying him (Eagleton 1990):

> *"God is a spineless liberal given to hollow authoritarian threats, who would never have the guts to perform what he promises [ . . . ] the point of Jonah's getting himself thrown overboard is to force God to save him, thus dramatically demonstrating to him that he's too soft-hearted to punish those who disobey him [ . . . ] Jonah doesn't believe for a moment that Nineveh's suspiciously sudden repentance is anything of his own doing: it has been brought about by God, to save himself the mess, unpleasantness and damage to his credibility as a nice chap consequent on having to put his threats into practice [ . . . ] God would have spared the city even if Jonah had stayed at home; it's just that he needs some excuse to do so. [ . . . ] And if God just goes around forgiving everybody all the time, what's the point of doing anything? If disobedience on the scale of a Nineveh goes cavalierly unpunished, then the idea of obedience also ceases to have meaning. God's mercy simply makes a mockery of human effort."*

We see these thoughts prominently portrayed in Robert Frost's play "*A Masque of Mercy*" (Frost 1947). Frost uses the central figure, Jonah Dove, a name which offers a double allusion to the biblical character, whose name "Jonah" means "dove", as the lens through which to ponder God's mercy and concomitantly his apparent failure to uphold divine justice. According to Jonah Dove, God, in his role as the supreme and just deity, must punish the wicked as anything else would be a miscarriage of justice. Jonah states poignantly:

> *I've lost my faith in God to carry out*
> *The threats He makes against the city evil.*
> *I can't trust God to be unmerciful.*
> Later in the same play, Jonah Dove continues:
> *I refuse to be the bearer of an empty threat.*

*He may be God, but me, I'm only human:*
*I shrink from being publicly let down.*
*[ . . . ]*
*There's not the least lack of the love of God*
*In what I say. Don't be so silly, woman.*
*His very weakness for mankind's endearing.*
*I love and fear Him. Yes, but I fear for Him.*
*I don't see how it can be to His interest*
*This modern tendency I find in Him*
*To take the punishment out of all failure*
*To be strong, careful, thrifty, diligent,*
*Anything we once though we had to be.*

Throughout the play runs the question of divine justice versus mercy. For Jonah, God's mercy absolves the consequences of human sin. Some of the other characters, however, maintain that God has the right to give his mercy to whomever he pleases. It is always freely given and cannot be earned (Loreto 1999, p. 29).

As already noted above, the book of Jonah is open-ended to the last. The readers will never know whether Jonah has become convinced by God's arguments and concedes that mercy is preferable to justice, or whether Jonah persists in his view that justice is better than mercy. Frost's play likewise ends without clear resolution, which forces the viewers to contemplate the repercussions of both options. Jonah dies, but it is up to the viewer to decide whether Jonah has accepted God's perspective or not (Timmerman 2002, p. 88).

Other retellings take a more explicit stand. In his novel *Profeten Jonas Privat* (Tandrup 1937), written in 1935 as a satire of Nazi Germany and its hatred of the Jews, the Danish author Harald Tandrup (1874–1964), envisages a Jonah who ultimately rejects God's perspective. Moreover, he regards Jonah's new standpoint as a positive development. Tandrup ends his retelling with Jonah standing up to God. Jonah has witnessed how the Assyrian monarch, incited by his priests, blamed the Jews for the assassination of his predecessor and has decided that it is best for the whole country if he commands the Jews to be killed. Jonah manages to survive the massacre, but his faith and trust in God, and his justice, is lost. Jonah does not see the point of believing in God, when, from Jonah's perspective, God has failed to uphold his part of the bargain. Jonah does not want to believe in a God that allows evil to exist unhampered in the world.

### 3.2.2. Jonah's Failure to Be Compassionate

Other retellings make the very opposite point and lament Jonah's failure to be compassionate. The musical rendering by the Jewish composer Samuel Adler (1928–), "*Der Mann ohne Toleranz*" ("The Man Without Tolerance") (Adler 2004), sung in German and Hebrew, laments Jonah's inability to rejoice at God's decision to pardon Nineveh. Using images of God speaking in the storm which echo God's decision to send the storm to thwart Jonah's attempt to flee (Jonah 1:4), as well as God's decision not to speak through the storm (1 Kings 19:11), Adler bemoans Jonah's closed mind. He refuses to hear God's side of the matter and see the justice in his concern for the innocent of Nineveh, who would have suffered had God not relented. Instead, Jonah walks away from the situation, unwilling and possibly also unable to relent:

*And Jonah went. And the burden of Nineveh that he had seen hung over his head, but he walked with a darkened mind. It howled in the storm and it cried in the wind and a voice cried out. For the sake of those, for the sake of those, for the sake of those, for the sake of those animals, clean and unclean. And the messenger of the Lord was stunned and looked but there was only darkness, and he heard nothing but an unrelenting howling and blowing that gripped his coat and pulled at it and shook it, like a pleader's hand [pulls] the garment of one unmercifully running away, but he did not relent; he strode and held his coat.* (Translation Andreas Tiemeyer)

3.2.3. Evaluation

Although these retellings are not the first to explore the questions of mercy and justice that the book of Jonah raises, they nonetheless constitute salient examples of reception exegesis insofar as they challenge us to contemplate, and often also to reevaluate, the traditional and well-established interpretation that lauds God's magnanimous mercy to the remorseful Ninevites.

Further, except for Adler's retelling, they invite us to side with Jonah and to make his questions our own. Jonah's statement in 4:2 shows his dissatisfaction with God's juridical posture: what God advocates is a situation where no equivalent suffering, no compensatory good deeds, and no expiatory rite, are needed for Nineveh to obtain forgiveness. As such, they question God's right to forgive an evil that was not done towards him but towards the victims of Nineveh. Jonah's initial flight from God in Jonah 1 and his concluding dialogue with God in Jonah 4 thus become examples of the human struggle against injustice. As aptly phrased by Yvonne Sherwood, the author of the book of Jonah may, in fact, try out the principle of universal mercy in the most extreme circumstances, asking a fantastic "What if" (Sherwood 1998, p. 67):

> What if Nineveh, the "bloody city" as Nahum puts it, the equivalent of Berlin of the Third Reich, repents?

These retellings further problematize the genre of the book. Rather than seeing it as a story of repentance and mercy (cf. above), they force readers to ponder whether the book of Jonah is not better read as a philosophical treatise on the essence of justice (cf. Rosen 1992). In my view, partly inspired by these retellings, the book of Jonah is best read as a post-exilic "historical novel" inspired by the reference to Jonah in 2 Kgs 14:23–25. From the eighth-century perspective presupposed in the story-world of Jonah, Nineveh, the capital of the Neo-Assyrian Empire, would eventually destroy the Northern Kingdom of Israel, as expounded in the book of Nahum. Reading Jonah and Nahum back-to-back, it is reasonable to conclude that the world would have been a better place had Jonah succeeded in his flight and God been able to destroy Nineveh unimpeded (Tiemeyer 2017).

## 4. Conclusions

The book of Jonah is a multifaceted book that raises many questions. The book is about God's calling, first to Jonah but, by extension, also to us, its readers. In the same way, the book is about God's forgiveness of undeserving sinners, again, first to Nineveh but, likewise, also to us. This dual message evokes strong emotional reactions. Yet, again like Jonah, it is our task as humans to struggle with it.

The cited twentieth-century literary retellings of the book of Jonah all deal with these fundamental issues. To return to Joyce and Lipton's words about reception exegesis that I cited at the beginning of this article, the retellings put a spotlight on theological and philosophical issues that are present in the text, but that are difficult to detect at first because we think that we know what the book of Jonah is all about. They challenge us to see beyond the obvious surface meaning, to question the overt interpretations of the Jonah story that we have heard time and again, and to stare our own unease with the interconnected fates of Jonah and Nineveh squarely in the face. Is Jonah right to flee from his calling and to question God's (from his perspective, misplaced) mercy, or is God right to insist that compassion should always triumph over justice, no matter the crime and no matter the personal cost?

On this note, I shall end with Dietrich Bonhoeffer's poem "*Jonah*", composed in his prison cell on 5 October 1944 (Bonhoeffer 1944). It is, in fact, possible that Bonhoeffer wrote this poem at the very moment when he gave up hope of escape and instead accepted the likelihood of death (Plant 2013, p. 66).

> [ . . . ] And Jonah spoke: "'Tis I!"
> In God's eyes I have sinned. Forfeited is my life.
> "Away with me! The guilt is mine. God's wrath's for me.

*The pious shall not perish with the sinner!"*
*They trembled much. But then, with their strong hands,*
*they cast the guilty one away. The sea stood still.*

This poem, with stark allusion to Jonah 1:16 and Jonah's prospect of a watery grave, is a fitting concluding example of reception exegesis. Again, it forms a two-way movement. Looking at the ways that the biblical echoes enhance Bonhoeffer's message, Solomon Liptzin argues that Bonhoeffer identifies with Jonah. Rather than being callous to the plight of others, Bonhoeffer, like Jonah, accepts his own responsibility for their situation and thus also the necessity of his own death (cf. Liptzin 1985, pp. 242–44). Along similar lines, Stephen Plant maintains that Bonhoeffer, like Jonah, confesses his sin and places his life humbly before the judgement of God, so that God alone can still the storm and calm the sea (Plant 2013, pp. 60, 69–70). At the same time, looking at how Bonhoeffer's poem enriches the interpretation of the biblical text, Bonhoeffer's poem highlights Jonah's moment of true heroism. Rather than allowing the sailors to die because of him, Jonah firmly gives them permission to throw him overboard. In a single, altruistic act, Jonah spares them from the crime of murder and instead mitigates it to assisted suicide. Thus, Jonah, so often maligned for his lack of mercy, shows true compassion for his fellow human beings.

**Funding:** This research received no external funding.

**Data Availability Statement:** Not applicable.

**Acknowledgments:** This article originated as a paper presented at the Winter Meeting of the British Society for the Old Testament Study in January 2020. I am much obliged for the feedback that I received on that occasion. I am also thankful for the constructive feedback from the two anonymous reviewers of this article. Much of the material in this article appears throughout my commentary *Jonah Through the Centuries*, albeit in a different form. I am furthermore applying the same methodology in my forthcoming book *In Search of Jonathan* (OUP). As a result, there is significant overlap, both in wording and in the cited scholarship, between the methodological discussion here and chapter 1 of that monograph.

**Conflicts of Interest:** The author declares no conflict of interest.

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
