# Peer review of "Jonah in 20th Century Literature"

_religions, doi:10.3390/rel13070661_

Round 1

Reviewer 1 Report

This is a fascinating and extremely well-executed article. The parallels between Jonah and Cain, in particular, are revelatory, and a superb example of reception exegesis. This section alone justifies publication, but in fact I learned so much more. One trivial type-setting comment: line 465 is the author's voice, not Frost's, and should not be indented or italicized. Other than that, I have nothing to add that could improve this outstanding contribution to the field.  

Author Response

Thank you so much for your encouraging feedback! It is much appreciated.

Reviewer 2 Report

The paper is well-written and adequately researched. The discussion includes a nice selection of 20th-century novels, poems and short-stories, which is both diverse and relevant to the general argument. At the same time, the discussion of these writings remains rather brief and could easily be extended, especially as regards their contribution to our understanding of the biblical story of Jonah,.

One potential issue concerns the concept of “reception exegesis,” which the author uses in this article. According to the initial statement, reception exegesis should help us identify issues that are present in the text, but are not correctly identified by the readers because of their own “predetermined ideas” about the biblical story. In other words, reception history—in this case literary reception—is used to challenge or even deconstruct what H.R. Jauß and others called the “horizon of expectation” (Erwartungshorizont). In and of itself, this is certainly a valid and productive approach. However, many of the examples discussed by the author are not so much about how modern literature can help us identify new issues or themes in the book of Jonah—in fact, most or even all of the themes discussed have long been identified as key topics of the book—but rather about exploring how modern authors have creatively reused these themes in their own writings. For instance, when dealing with the topic of Jonah fleeing away from his call, the author concludes: “…the retellings change our appreciation of the book of Jonah insofar as we may feel compassion for Jonah. The biblical prophet … is transformed into a man of flesh and blood that we can relate to” (5). The point here, and in other examples as well, is not that modern literature helps us identifying new questions or themes in the biblical account of Jonah, but rather that the retellings allow the reader to access an alternative account of the biblical story which is (perhaps) more attuned to our sensibilities and interrogations. Again, all the themes of Jonah discussed by the author are not particularly difficult to detect, and have long been discussed.

The author only provides one significant example where reading a modern novel leads to asking new questions about the biblical story itself, and it is rather convoluted: the author claims that reading the novel by Moacyr Scliar (The Strange Nation of Rafael Mendes) led him or her to study the connections between Jonah and Genesis 1-4, but why this is so is never really explained. In this reader’s opinion, the way in which the writings discussed in this article contribute to the method of “reception exegesis” could be better clarified.

Two remarks of detail:

- on p. 2 the author introduces Meir Steinberg’s distinction between “legitimate” and “illegitimate” gap-filling, before moving on to compare modern retellings with midrash. However, authors of midrash, whether ancient or modern, would usually not regard their own retelling of the biblical story as “illegitimate” (in a sense, it is often the point of these retellings that the author feels he or she has something legitimate to say about these stories!), and it is far from clear what role this cocnept plays in the following discussion. The whole paragraph is very general and could easily be shortened or even omitted.

- p. 11-12: as much as Dietrich Bonhoeffer’s poem cited in the end is poignant, one wondersy what point exactly the author is trying to make? Also what should the reader make of the final statement according to which this poem, which the author admits not fully understanding, “offers a fitting conclusion to this article”? This is rather confusing.

All in all, this is an interesting article, which definitely has some potential. In its present state, however, it still looks more like an exploration of the creativity of 20th-century writers in their various retellings of Jonah than a systematic application of the method of “reception exegesis,” as the article claims to be. In addition, one may also ask whether this article really belongs to a journal devoted to the study of religions, or whether it would not be better placed in a journal for literature or reception history. To be sure, the author speaks of the Bible and of God, but in terms of the study of religion as an academic discipline, the contribution seems rather limited. At the very least, it would be helpful that the author explains how s/he views the article’s contribution to the study of religion.

Author Response

Thank you for your feedback. It has helped me clarify some aspects of my writing and thus hopefully resulted in a more tightly edited and better-focused article.